# Vitamin D Supplementation Reduces Both Oxidative DNA Damage and Insulin Resistance in the Elderly with Metabolic Disorders

**DOI:** 10.3390/ijms20122891

**Published:** 2019-06-13

**Authors:** Sylwia Wenclewska, Izabela Szymczak-Pajor, Józef Drzewoski, Mariusz Bunk, Agnieszka Śliwińska

**Affiliations:** 1Department of Internal Medicine, Diabetology and Clinical Pharmacology, Medical University in Lodz, 251 Pomorska Street, 92-213 Lodz, Poland; sylwia.wenclewska@umed.lodz.pl; 2Department of Nucleic Acid Biochemistry, Medical University of Lodz, 251 Pomorska Street, 92-213 Lodz, Poland; izabela.szymczak@umed.lodz.pl; 3Central Teaching Hospital of the Medical University of Lodz, 251 Pomorska Street, 92-213 Lodz, Poland; jozef.drzewoski@umed.lodz.pl (J.D.); mariuszbunk@gmail.com (M.B.)

**Keywords:** vitamin D, endogenous and oxidative DNA damage, type 2 diabetes, lipid profile, insulin resistance

## Abstract

**Background:** Research evidence indicates that vitamin D deficiency is involved in the pathogenesis of insulin resistance (IR) and associated metabolic disorders including hyperglycemia and dyslipidemia. It also suggested that vitamin D deficiency is associated with elevated levels of oxidative stress and its complications. Therefore, the aim of our study was to determine the effect of vitamin D supplementation on DNA damage and metabolic parameters in vitamin D deficient individuals aged >45 with metabolic disorders. **Material and Methods:** Of 98 initially screened participants, 92 subjects deficient in vitamin D were included in the study. They were randomly assigned to the following group: with vitamin D supplementation (intervention group, *n* = 48) and without supplementation (comparative group, *n* = 44). The patients from both groups were divided into two subgroups according to the presence or absence of type 2 diabetes (T2DM). The intervention group was treated with 2000 International Unit (IU) cholecalciferol/day between October and March for three months. At baseline and after three-month supplementation vitamin D concentration (25-OH)D3 and endogenous and oxidative DNA damage were determined. In addition, fast plasma glucose (FPG), fasting insulin, HbA1c and lipid fraction (total cholesterol (TC), low-density lipoprotein cholesterol (LDL), high-density lipoprotein cholesterol (HDL), triglyceride (TG)), as well as anthropometric measurements (weight, height) were gathered. The following IR-related parameters were calculated Homeostatic Model Assesment – Insulin Resistance (HOMA-IR) and TG/HDL ratio. **Results:** Three-month vitamin D supplementation increased the mean vitamin D concentration to generally accepted physiological level independently of T2DM presence. Importantly, vitamin D exposure decreased the level of oxidative DNA damage in lymphocytes of patients of intervention group. Among studied metabolic parameters, vitamin D markedly increased HDL level, decreased HOMA-IR, TG/HDL ratio. Furthermore, we found that HbA1c percentage diminished about 0.5% in T2DM patients supplemented with vitamin D. **Conclusion:** The current study demonstrated that daily 2000I U intake of vitamin D for three months decreased the level of oxidative DNA damage, a marker of oxidative stress, independently on T2DM presence. Furthermore, vitamin D reduced metabolic parameters connected with IR and improved glucose and lipid metabolism. Therefore, our results support the assertion that vitamin D, by reducing oxidative stress and improving of metabolic profile, may decrease IR and related diseases.

## 1. Introduction

Vitamin D is a fat-soluble steroid hormone, made of a sterol ring, that together with calcitonin and parathyroid hormone plays a role as a main regulator of calcium and phosphate metabolism. The discovery that vitamin D receptors (VDR) are present not only in bone cell and kidney tubes, but also in the majority of human cells, was a breakthrough. As a result, growing evidence has reported about its extra-skeletal functions. Vitamin D was found to be involved in maintenance of glucose homeostasis through numerous mechanisms connected with the insulin signaling pathway. Vitamin D takes part in sustaining of normal level of Ca^2+^ and reactive oxygen species (ROS) both in pancreatic beta cells and insulin responsive cells. It is well recognized that augmented oxidative stress and vitamin D deficiency are associated with insulin resistance (IR)-related diseases, including type 2 diabetes (T2DM) [1,2,3,4]. 

A large number of studies support improving T2DM control, decreasing of insulin resistance, obesity, and metabolic syndrome with vitamin D adequacy. It was also found that vitamin D supplementation improved glycemic control and attenuated insulin resistance in prediabetics or individuals at high risk of diabetes development [5]. Beyond the impact on the glycemic parameters, observational studies showed that high level of vitamin D is associated with a favorable lipid profile, while low levels of vitamin D are connected to an atherogenic lipid profile [6]. Vitamin D deficiency and insufficiency is a global health issue afflicting more than one billion children and adults worldwide [7]. In Poland, 90% of healthy adults have vitamin D concentrations below 30 ng/mL. An epidemiological study conducted in 2016 revealed that mean vitamin D serum concentration in the Polish population was 18.0 ± 9.6 ng/mL; 65.8% of the patients had a level below 20 ng/mL, 24.1% had suboptimal level of 20 to 30 ng/mL and only 9.1% demonstrated the optimal level of 30 to 50 ng/mL [8].

Therefore, the aim of our study was to evaluate whether three-month vitamin D supplementation is enough to increase its level to physiological concentration and to affect the level of endogenous and oxidative DNA damage being considered as a marker of oxidative stress. Moreover, we determined the effect of vitamin D on body mass index (BMI), metabolic parameters and IR indicators. 

## 2. Results 

### 2.1. Three-Month Supplementation with 2000 IU of Vitamin D/Day Increased Concentration of Vitamin D to the Level Regarded as Physiological

No significant difference was seen in vitamin D level at baseline between intervention and comparative group. However, after a three-month treatment, the level of vitamin D significantly increased (31.81 vs. 19.80, *p* < 0.005) as shown in Figure 1. In the group not supplemented with vitamin D, its serum level did not change significantly after three months. In the intervention group the concentration of vitamin D rose above 30 ng/mL. As presented in Figure 2, after three-month supplementation with 2000 IU of vitamin D, the T2DM group (T2DM + Vit. D) and the group without T2DM (Control + Vit. D) achieved its physiological level. In groups with T2DM and without T2DM not supplemented with vitamin D (T2DM − Vit. D and Control − Vit. D) the level of this vitamin was similar to the level detected at baseline (Figure 2). It should be noted that we did not find the statistically different level of vitamin D between T2DM − Vit. D vs. T2DM + Vit. D after three months, but still it was a trend toward increase in T2DM + Vit. D group. This could be the effect of relatively small number of subjects in T2DM group and variation of vitamin D level in particular subjects at time 0, especially in the T2DM − Vit. D group.

### 2.2. Three-Month Vitamin D Supplementation Decreased Oxidative DNA Damage

We evaluated the effect of three-month vitamin D supplementation on the level of endogenous and oxidative damage of DNA in lymphocytes (Figure 3). We observed that at baseline the level of endogenous and oxidative DNA damage was similar in lymphocytes in the intervention and comparative group. Three-month treatment with vitamin D evoked significant decline of the level of oxidative DNA damage (recognized by Fpg *p* < 0.05). We did not observe any changes in the level of endogenous and oxidative DNA damage (recognized by Nth) in lymphocytes of intervention group. The level of endogenous and oxidative (recognized by Fpg) DNA damage did not change after three months in lymphocytes of patients in the comparative group. Surprisingly, the level of oxidative DNA damage recognized by Nth increased after three months in lymphocytes of patients not exposed to vitamin D (*p* < 0.01). 

### 2.3. Three-Month Vitamin D Supplementation Decreased the Level of Endogenous and Oxidative DNA Damage in Lymphocytes of Type 2 Diabetes (T2DM) Patients

The analysis of DNA damage performed in T2DM groups revealed that at the beginning of the study the level of endogenous DNA damage was higher in both T2DM in comparison to both control group (Figure 4). Similarly, the level of oxidative DNA damage (recognized by FPG) was higher than endogenous DNA damage in all studied groups. We found that three-month vitamin D supplementation (T2DM + Vit. D) caused a pronounced diminish of the level of endogenous and oxidative DNA damage in lymphocytes of T2DM patients, while in lymphocytes of T2DM patients not supplemented with vitamin D (T2DM − Vit. D) the level of endogenous and oxidative DNA damage (recognized by FPG) did not change significantly. Interestingly, we noticed that after three months there was a significant increase in the level of oxidative DNA damage (recognized by Nth, *p* < 0.01) in lymphocytes of T2DM patients not supplemented with vitamin D (T2DM − Vit. D). In the control group (Control + Vit. D) three-month vitamin D supplementation evoked marked decrease of oxidative DNA damage (recognized by Fpg, *p* < 0.05). No significant changes of studied DNA damage were found in lymphocytes of control group not supplemented with vitamin D (Control − Vit. D).

### 2.4. Three-Month Supplementation Increased the Level of High-Density Lipoprotein Cholesterol (HDL), and Decreased HOMA-IR and Triglyceride (TG)/HDL Ratio

The effect of vitamin D supplementation on selected metabolic parameters such as FPG, TC, HDL, LDL, TG, HbA1c, BMI, HOMA-IR and TG/HDL ratio was evaluated. After three months of supplementation we did not observe any effect of vitamin D on FPG, TC, LDL, TG and HbA1c level in the intervention group. It should be emphasized that the intake of 2000 IU vitamin D/day led to a significant increase in HDL (*p* < 0.001) and pronounced decrease in HOMA-IR (*p* < 0.005) and TG/HDL ratio (*p* < 0.005) in the intervention group. Otherwise, after three months in the intervention group, a trend toward an increase of TC and LDL was noted. In the comparative group, after three months, a significant reduction of FPG and HOMA-IR were observed as displayed in Table 1. 

Considering the analysis in group with and without T2DM, we found that three-month supplementation with vitamin D markedly increased level of HDL and decreased TG/HDL ratio in both groups (T2DM + Vit. D and Control + Vit. D). It is worth noting that the TG/HDL ratio diminished in all four groups, but in groups with vitamin D it was statistically significant. These results may suggest that vitamin D could improve the lipid profile by increasing HDL level, and in consequence a reduction of the TG/HDL ratio. We also observed a decrease in HOMA-IR in groups supplemented with vitamin D (T2DM + Vit. D and Control + Vit. D). Interestingly, we reported HOMA-IR reduction in the control group without supplementation (Control − Vit. D). The decrease of HOMA-IR found in the control group without vitamin D supplementation (Control − Vit D) might be a result of lifestyle changes recommended to patients who at baseline had FPG level between 100–125 mg/dL considered as prediabetic state. These recommendations included the use of diet with a reduced calorie content, physical activity (150 min/week) which resulted in weight loss and diminished BMI. As a result in this control group (Control − Vit. D) we found significantly lower FPG after three months, which markedly decrease the mean value of HOMA-IR. 

Similarly to the intervention group, after three months of supplementation with vitamin D, we found a significant increase of TC and a trend toward increase in LDL in T2DM + Vit. D group. The trend toward increase level of TC and LDL-C was also observed in group without T2DM supplemented with vitamin D (Control + Vit. D). Elevated level of TC in T2DM participants supplemented with vitamin D could be a result of a pronounced growth of HDL concentration (*p* < 0.001). We found that vitamin D did not significantly affect TG level. We observed a trend toward a decrease in HbA1c percentage (about 0.5%) in the T2DM group after supplementation with vitamin D (T2DM + Vit. D), Table 2. The trend towards lowering HbA1c in patients with T2DM may suggest a beneficial effect of this vitamin on metabolic control of the disease.

### 2.5. The Correlation Analysis Supported the Assertion That Three-Month Vitamin D Supplementation Reduced Oxidative DNA Damage and Improved Metabolic Control of T2DM

To check whether vitamin D may affect endogenous and oxidative DNA damage we performed Pearson correlation analysis as shown in Table 3 and Table 4. We did not find any significant correlation between vitamin D and endogenous and oxidative DNA damage at baseline and after three-month supplementation, both in the intervention and comparative groups. A trend toward negative correlation between vitamin D and oxidative DNA damage (recognized by Nth) after three-month supplementation in the intervention group was observed (Table 3). This result may imply that with increasing level of vitamin D, the level of oxidative DNA damage decreases. The analysis performed in groups with and without T2DM is presented in Table 4. We did not reveal any significant correlation between vitamin D level and endogenous as well as oxidative DNA damage at baseline and after three months of vitamin D supplementation, both in patients with and without T2DM.

As a HbA1c is the marker of metabolic control of diabetes, the correlation between HbA1c and endogenous and oxidative DNA damage in the group of patients with T2DM (+Vit. D and −Vit.D) was also performed, which is presented in Table 5. We found a statistically significant positive correlation between HbA1c and oxidative DNA damage (recognized by Nth) at baseline in T2DM+Vit. D. After three months of supplementation we did not find a significant correlation between HbA1c and endogenous and oxidative DNA damage, in the T2DM group supplemented with vitamin D. This observation supports the suggestion that vitamin D may have a beneficial effect on metabolic control of T2DM.

### 2.6. Three-Month Vitamin D Supplementation Diminished HbA1c Percentage

In order, to evaluate the impact of vitamin D on metabolic parameters we performed a correlation analysis using Pearson correlation test as presented in Table 6. At baseline, we did not observe any significant correlation between vitamin D level and FPG, TC, LDL, TG, HbA1c, BMI, HOMA-IR and TG/HDL ratio both in the intervention and comparative group. However, we found that vitamin D deficiency was positively correlated with lower HDL level in both groups, although it was only statistically significant in the intervention group. This suggests that vitamin D deficiency may be associated with lower HDL level. After three months of vitamin D supplementation, we still did not report any significant correlation between vitamin D level and metabolic parameters in the comparative group. In turn, in the intervention group after three months we found a weak negative correlation between vitamin D and HbA1c percentage. This observation suggests that with increasing vitamin D level the percentage of HbA1c decreases. 

Since we observed a statistically significant effect of vitamin D supplementation on HbA1c percentage we decided to conduct Pearson correlation analysis in groups with and without T2DM. Because of no proven clinical value of HbA1c measurement in healthy population, we did not perform this correlation analysis in patients without T2DM. The correlations between vitamin D and metabolic parameters in T2DM groups (+Vit. D and −Vit. D) and control groups (+Vit. D and −Vit. D) are shown in Table 7. At baseline there were no statistically significant correlations between vitamin D and the most of the studied parameters, except TG level and TG/HDL ratio in T2DM + Vit. D group. It should be emphasized that at baseline we did not observe any correlation between vitamin D and HbA1c in T2DM groups, but after three-month vitamin D supplementation, we found a strong negative correlation between these parameters in the T2DM + Vit. D group. Analysis of the correlation between vitamin D and BMI in groups with and without T2DM, showed no relationship between these parameters at baseline. We found a significant negative correlation between vitamin D and BMI in group without T2DM supplemented with vitamin D (Control + Vit. D) after three months and trend toward this correlation in T2DM + Vit. D group. It may suggest that achieving physiological concentration of vitamin D may facilitate a reduction in weight and maintain BMI.

## 3. Discussion 

The primary function of vitamin D is the regulation of bone metabolism and calcium-phosphate homeostasis. Also, studies have reported that vitamin D may play an important role in the maintenance of pancreatic beta cell function [1]. It is achieved by the activation of the vitamin D receptor (VDR) and regulation of insulin secretion via the calcium channel located in pancreatic beta cells. A deficiency of vitamin D was found to be be associated with the development of various diseases, including IR-dependent diseases such as T2DM, obesity, dyslipidemia, cardiovascular diseases (CVDs) [9]. Growing body of evidence shows that oxidative stress plays an important role in the pathogenesis of IR-dependent disorders. It suggests that vitamin D can reduce deleterious effect of oxidative stress [10]. Thus, in this study we hypothesized that supplementation of vitamin D can reduce the level of oxidative DNA damage. 

Our result showed that, vitamin D decreased oxidative DNA damage. Our results are in line with Fedriko et al. and Lan et al. [11,12]. We demonstrated that vitamin D supplementation decreased the level of endogenous and oxidative DNA damage in lymphocytes of T2DM patients after three months. These results confirm that vitamin D may reduce the adverse effects of oxidative stress in patients with diabetes. It is well known that diabetes via chronic hyperglycemia is associated with an increased level of oxidative damage, including DNA damage [13]. What is more, high percentage of HbA1c detected in T2DM patients reflects high glucose level during the last 2–3 months. Thus, high percentage of HbA1c corresponds to poor glycemic control, hyperglycemia and related oxidative stress. Both the increased DNA damage and HbA1c percentage in our study at baseline suggest that patients with T2DM were poorly controlled and developed oxidative DNA damage associated with oxidative stress. Additionally, we observed that the increase in vitamin D concentration was associated with a reduction in HbA1c percentage in T2DM patients. This observation supports the beneficial effects of vitamin D on the metabolic control of diabetes which can be seen as a decrease in insulin resistance. These findings suggest that vitamin D may contribute to better diabetes alignment by reducing oxidative stress. In practice, this effect may translate into a delay in the occurrence of diabetes complications, including micro- and macroangiopathies. 

In this study lipid profile, glucose homeostasis parameters (FPG, HbA1c), BMI, IR markers (HOMA-IR, TG/HDL ratio) and the level of endogenous and oxidative DNA damage were measured at baseline and after three-month supplementation with 2000 IU vitamin D/day. We found a significant effect of three-month vitamin D supplementation on HDL level. The increase in HDL was noted in the intervention group, but its growth was particularly visible in the T2DM group (T2DM + Vit. D). This effect of vitamin D may be especially important for patients with T2DM who frequently suffer from obesity-related dyslipidemia. It is primarily characterized by increased levels of plasma free fatty acids and TG, decreased levels of HDL and abnormal low-density lipoprotein (LDL) composition. Additional benefit can be achieved by patients who are treated with statins, since these drugs do not affect HDL level significantly. The advantageous effect of vitamin D on HDL was also confirmed by the correlation analysis that showed a significant positive correlation between vitamin D and HDL at baseline in the intervention group. It suggests that vitamin D deficiency is correlated with the low level of HDL cholesterol. This observation stays in accordance with the study by Jiang et al., who detected vitamin D deficiency correlated with low HDL level in 3788 Chinese adults aged 35–74 [14]. When, we performed the correlation analysis after three-month vitamin D supplementation, the relationship between vitamin D and HDL was not observed. It supports that vitamin D has a beneficial impact on HDL level. This desired effect of vitamin D on HDL was also confirmed in the meta-analysis by Mirrhosseini et al. [15]. 

In turn, our results concerning the impact of vitamin D on other metabolic parameters are ambiguous. We did not find any significant effect of three-month vitamin D supplementation on TC, LDL and TG in the intervention group. On the one hand, our results are consistent with previous studies showing that one-year supplementation with 20,000 or 40,000 IU/week vitamin D did not improve serum lipids in Caucasian subjects [16]. Grimnes et al. showed that six months of 40,000 IU/week vitamin D treatment did not exert a beneficial effect on lipid profile [17]. On the other hand, the results of the present study are in contrast to the findings of Qin et al. [18] who observed metabolic changes, including improvement of TC and TG levels evoked by vitamin D supplementation. It should be emphasized that this study had a longer follow-up period compared to our study, although the dose of vitamin D was the same. Similarly, Barzegari et al. achieved a significant reduction of TC, LDL and TG level after vitamin D supplementation. The study was conducted on 25 patients deficient in vitamin D with T2DM, who received this vitamin for eight weeks at a dose of 50,000 IU/week [10]. In the face of ambiguous results regarding the effect of vitamin D on lipid profile, especially in metabolic disorders, further research is needed.

In our study, we also evaluated the effect of three-month vitamin D supplementation on glucose homeostasis parameters (FPG, HbA1c). We found a significant decrease of FPG in the comparative group, T2DM + Vit. D group and Control − Vit. D group. The reduction of FPG in T2DM group with vitamin D supplementation (T2DM + Vit. D) stays in accordance with meta-analysis by Mirrhosseini et al. [5]. They found that vitamin D supplementation caused a significant decrease of FPG in diabetes patients. The significant decrease in FPG in control group without supplementation (Control − Vit. D) might be caused by the influence of other factors. It is worth noting that this group was characterized by a high level of FPG at baseline suggesting the presence of patients with prediabetes. The changes of lifestyle such as moderate physical activity and healthy diet were recommended for these participants. As a result of these changes, we observed a decrease in HOMA-IR and a trend toward decrease in TG/HDL ratio and BMI in the Control − Vit. D group. There was no significant diminish in HbA1c percentage in the intervention group, but in T2DM patients (T2DM + Vit. D) we found a trend toward decrease of HbA1c percentage (0.46 %). It is generally accepted that lower HbA1c, results in lower risk of diabetes-related complications [19]. This desired effect in T2DM patients (T2DM + Vit. D) was also confirmed by the correlation analysis. It showed a significant negative relationship between vitamin D and HbA1c percentage after three-month vitamin D supplementation. However, other studies on the effect of vitamin D on HbA1c are inconclusive. Al-Sofiani et al. demonstrated that HbA1c did not change considerably after intervention of 5000 IU/day of vitamin D for 12 weeks [20]. Likewise, Sadyia et al. also reported no significant change in HbA1c percentage after six months of supplementation with vitamin D of Emirati population [21]. The study conducted by Tabesh et al. [22] stays in agreement with our results. It demonstrated a decrease in HbA1c percentage in volunteers using 50,000 IU/week of vitamin D in combination with 1000 mg/day of calcium for six months. Upreti et al. found that six-month vitamin D supplementation was associated with a decrease in HbA1c percentage in patients with T2DM [23]. Altogether, the findings suggest that vitamin D may exert a favorable effect on diabetes control.

Due to fact that IR is one of the major hallmark of T2DM [24], we analyzed the effect of vitamin D on IR markers, including HOMA-IR and TG/HDL ratio. McLaughin et al. were first to report the potential utility of the TG/HDL ratio to detect IR in Caucasian population [25]. We found that three-month vitamin D supplementation significantly reduced TG/HDL ratio and HOMA-IR in the intervention group. The decrease in HOMA-IR was also observed in the comparative group. Analyzing groups with and without T2DM, we noticed a statistically significant decrease of both IR indicators in groups supplemented with vitamin D (T2DM + Vit. D and Control + Vit. D). This is in line with findings by El Haji et al. who observed significant decrease in HOMA-IR in old adults supplemented with 4000 IU/day of vitamin D for 6 months [26]. Similarly, Mirrosheini et al. showed that vitamin D diminished HOMA-IR [27]. Taken together, these findings strongly support that vitamin D is effective in IR reducing. It should be also noted, that we observed a significant inverse correlation between vitamin D and BMI in patients without T2DM (Control + Vit. D group) and a trend toward this correlation in the T2DM + Vit. D group. This may suggest that higher vitamin D concentration may facilitate the reduction of weight and maintenance of proper BMI. This result stays in line with observation of Cassity et al. who reported that after monthly supplementation with 4000 IU/day of vitamin D BMI value was diminished [28].

The study was limited by a relatively small research group, especially T2DM subgroups. As a result, the analysis adjusted to gender was not performed. The strength of the study is the analysis of metabolic and anthropometric parameters. Our results enhance to extend the supplementation period to the whole autumn and winter season, especially in the group of patients with T2DM, who may benefit more from the supplementation with vitamin D.

## 4. Material and Methods

### 4.1. Vitamin D

Vitamin D (Symvitum D3) was provided by Symphar, Warsaw, Poland One capsule contains 2000 IU of cholecalciferol suspended in high-quality MCT (medium-chain triglicerydes) oil.

### 4.2. Study Design

The study protocol was approved by the Independent Bioethics Committee of the Medical University of Lodz (file number: RNN/223/15/KE 06.07.2015; approval date: 6 July 2015, and conducted in accordance with good Clinical Practice and the Declaration of Helsinki. All participants provided written informant consent before enrollment to the study. Inclusion criteria were: age above 45 years, vitamin D level lower than 30 ng/mL. Participants with vitamin D serum concentration higher than 30 ng/mL, younger than 45 years of age, suffering from diabetes other than T2DM, treated with insulin, sulphonylurea drug, fibrate and blood products transfusions were excluded from the study. Of 98 initially qualified participants, 92 were included in the study. Participants were recruited from the Department of Internal Medicine, Diabetology and Clinical Pharmacology in Lodz, Poland, between October 2016 to March 2018. Supplementation with vitamin D began from 1 October to 31 December. Eligible participants were then randomized to intervention group (*n* = 48) and to comparative group (*n* = 44) as presented in Figure 5. The intervention group received 2000 IU of vitamin D3 (cholecalciferol) per day (one capsule 2000 IU each) for three months, from October to March. At baseline and after the completion of vitamin D supplementation, vitamin D concentration, fast plasma glucose (FPG), fasting insulin, HbA1c, the level of total cholesterol (TC), triglyceride (TG), low-density lipoprotein cholesterol (LDL), high-density lipoprotein cholesterol (HDL) were determined. Weight and height were also gathered. Using the obtained data, HOMA-IR, TG/HDL ratio and BMI were calculated.

### 4.3. Baseline Characteristics of Patients

The intervention group consisted of 34 women and 14 men, while in the comparative group there were 25 women and 19 men. The mean age of the intervention group was 63.43 ± 1.57 years and 69.78 ± 2.10 of comparative group. Metabolic characteristics of patients included to the studied groups are presented in Table 1. According to acquired data, there were no statistically significant differences in metabolic parameters between intervention and comparative group at baseline, apart from TG (149.13 ± 12.64 vs. 117.15 ± 8.56, *p* < 0.05), TC (189.54 ± 7.94 vs. 162.12 ± 7.34, *p* < 0.05) and HOMA-IR (5.55 ± 0.59 vs. 7.85 ± 0.89, *p* < 0.05). These differences were probably associated with age, T2DM presence, differences in body weight, and lipid-lowering drugs usage. Eighteen participants in the intervention group (18/48) and 14 in the comparative group (14/44) had T2DM reported by participants or were classified in the T2DM group based on the usage of hypoglycemic medication before study begun. The mean age of T2DM group supplemented with vitamin D (T2DM + Vit. D) was 69.78 ± 2.19 years and 72.57 ± 3.02 years in T2DM group without supplementation (T2DM − Vit. D). In the studied groups without T2DM, the mean age was 59.8 ± 1.83 years in group with vitamin D supplementation (Control + Vit. D) and 68.33 ± 2.71 years in group without supplementation (Control − Vit. D). The metabolic characteristic of participants adjusted to T2DM presence is depicted in Table 2. At baseline, there were no statistically significant differences in metabolic parameters between T2DM + Vit. D group and T2DM − Vit. D group. In the groups without T2DM (Control + Vit. D vs. Control − Vit. D) statistically significant differences in TC, TG levels, BMI and HOMA-IR were found. Both TC and TG level were significantly higher at the beginning of the study in the control group with vitamin D supplementation (Control + Vit. D), while FPG level, BMI and HOMA-IR was significantly lower in the above group. It could be related to the fact that this group was younger than other groups, participants did not report the use of drugs affecting lipid metabolism parameters and they are also had lower body weight that affected BMI value. (75.77 ± 2.00 vs. 83.99 ± 1.29 kg). There were no significant differences in other tested parameters i.e., HDL, LDL, HbA1c, TG/HDL ratio between control groups (Control + Vit.D vs. Control − Vit. D). Both T2DM groups (T2DM + Vit. D and T2DM − Vit. D) were significantly older and had significantly higher FPG, HbA1c and TG/HDL ratio in relation to groups without diabetes (Control + Vit. D and − Vit. D) at baseline. Also, lower TC and LDL levels were found in both T2DM groups (T2DM + Vit. D and T2DM − Vit. D) in comparison to both control groups (Control + Vit. D and − Vit. D). This observation was a result of the use of lipid-lowering drugs—statins, by participants with diabetes. In addition, lower HDL level was noted in the groups with T2DM (T2DM + Vit.D/− Vit. D). This metabolic profile corresponds to atherogenic dyslipidemia which is a typical feature of T2DM. According to the definition, it is characterized by the coexistence of decreased HDL and increased TG concentration. We also found higher TG level in both groups of patients with T2DM (T2DM + Vit. D/− Vit. D) in comparison with the control group without vitamin D supplementation (Control − Vit. D), but not in comparison to Control + Vit. D group.

### 4.4. Methods

The serum vitamin D level was determined using the CLIA method on Liaison analyzer (Diasorin, Saluggia, Italy) in accordance with the manufacturer instruction. The reference range of vitamin D concentration was established as: lower than 10 ng/mL—insufficiency, 10–30 ng/mL—inadequately low, 30–100 ng/mL—sufficient, more than 100 ng/mL—toxic. A blood sample (15 mL) was collected in the morning after at least eight hours of overnight fasting. Serum levels of FPG, HbA1c, TC, HDL, LDL, TG were assessed using standard analytical methods. HbA1c level was determined by turbidimetric immunoinhibition method, FPG level was photometrically measured in ultraviolet light using enzymatic reactions catalysed by hexokinase. The enzymatic colorimetric test was quantified for TC, HDL and TG using Beckman Coulter AU analyzers (Beckman Coulter, Brea, CA, USA). LDL was calculated using the Friedewald formula. Insulin concentration was determined by electrochemiluminescence test. HOMA-IR and TG/HDL ratio were calculated from the results obtained. 

#### 4.4.1. The Alkaline Comet Assay

Venous blood samples were collected from each participant before breakfast at the baseline and after three-month supplementation of vitamin D. Lymphocytes of peripheral blood were isolated by centrifugation in a density gradient using Histopaque (Sigma Aldrich, Munich, Germany) and washed three times with phosphate buffered saline (PBS). The alkaline version of comet assay was performed according to the protocol of Singh et al. [29] with previously described modifications [30,31]. This comet assay version detects single and double strand breaks as well as the alkaline-labile sites, which are considered endogenous DNA damage. Isolated lymphocytes were suspended in low melting point agarose (0.75%) and spread onto normal melting point agarose (0.5%) precoated microscope slides. Then, the cells were lysed by incubation in lysis buffer (NaCl, 2.5 M; EDTA, 100 mM; TritonX100, 1%; and Tris, 10 mM; pH 10) at 4 °C for 1 h. Next, the slides were suspended in unwinding buffer (NaOH, 300 mM; EDTA, 1 mM; pH >13) and electrophoresis was carried out at 0.73 V/cm (28 mA) for 20 min. After electrophoresis the slides were washed three times with distilled water, drained, as well as being stained with 2 mg/mL 4′.6diamidino2phenylindole dihydrochloride (DAPI) under dark conditions at a temperature of 4 °C for 30 min. Finally, the comets were seen under a fluorescence microscope (Nikon, Tokyo, Japan) at a magnification of ×200 connected to a video camera with ultraviolet (UV1), a filter block and personal computer equipped with the LuciaComet v. 4.51 analysis software (Laboratory Imaging, Prague, Czech Republic). DNA damage as the percentage of DNA in the tail of the observed comet was evaluated from 50 cells in each sample.

#### 4.4.2. DNA Repair Enzyme Treatment 

Formamidopyrimidine DNA glycosylase (Fpg) and endonuclease III (Nth) were used to determine the level of oxidative DNA damage. Fpg is an enzyme playing a crucial role in the first step of base excision repair. This enzyme cuts and removes mainly 7,8-dihydro-8-oxo-2`deoxyguanine (8oxoG) and 2,6-diamino-4-hydroxy-5-N-methyl formamidopyrimidine from DNA leading to the generation of apurinic/apyrimidinic sites [32,33,34]. Nth is a restriction endonuclease responsible for the identification of oxidized pyrimidines and their transformation into DNA strand breaks detected by the comet assay [34]. To study the capability of the enzymes to recognize oxidized bases, the lymphocytes were lysed and incubated with Nth and Fpg enzymes. The slides with cells were washed in enzyme buffer (HEPES-KOH, 40 mM; KCl, 0.1 mm; EDTA, 0.5 mM; bovine serum albumin, 0.2 mg/mL; pH. 8.0), drained and 25 µl of a mixture of enzyme buffer with 1 µg/mL of the enzyme or only enzyme buffer was placed on slides. Then, the slides were covered by cover slips and incubated at 37 °C for 30 min [34,35]. After incubation, the cells were suspended in unwinding buffer and electrophoresis was carried out. To analyze the net value of DNA damage recognized by the enzymes, we subtracted the DNA damage observed in the absence of the enzymes from that measured in the presence of them.

#### 4.4.3. Data Analysis

In case of data derived from the comet assay, the results are presented as mean and standard error of the mean (±SEM). If no significant differences between variations assessed by the Snedecor–Fisher test and the distributions of variables were determined as in accordance with normal by the Shapiro–Wilk’s test then the differences between the means were assessed using the *t* test. *p* value less than 0.05, was considered as statistically significant. All statistical analysis were performed using GraphPad Prism 6.0 (GraphPad, San Diego, CA, USA).

## 5. Conclusions

To conclude, our study revealed that supplementation with 2000 IU/day of vitamin D decreased the level of DNA damage and this effect was especially visible in T2DM patients. The reduction of DNA damage by vitamin D suggest that this vitamin may reduce oxidative stress resulting from hyperglycemia. It was also noted that vitamin D supplementation was associated with an increase in HDL level, and reduced HOMA-IR and TG/HDL ratios. Moreover, our results suggest that patients with T2DM may benefit more from supplementation with vitamin D than the general population, since it reduces not only IR indexes, but also HbA1c level. 

## Figures and Tables

**Figure 1 ijms-20-02891-f001:**
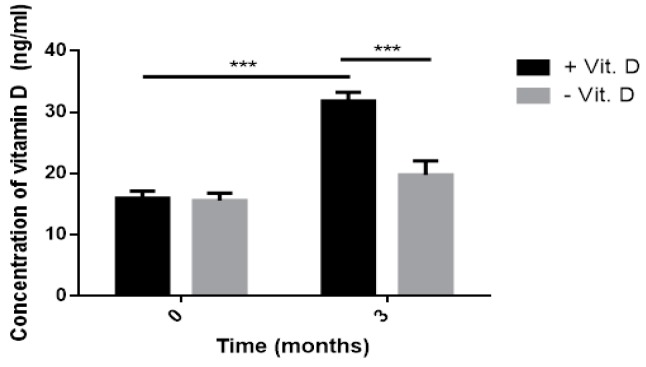
The serum level of vitamin D detected in the intervention group (+Vit. D, *n* = 48) and the comparative group (−Vit. D, *n* = 44) before and after three-month supplementation. Results are presented as mean ± standard deviation (SD) (*** *p* < 0.001).

**Figure 2 ijms-20-02891-f002:**
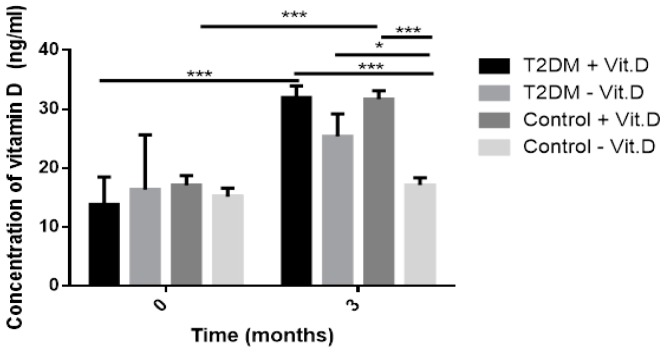
The serum level of vitamin D detected in patients with type 2 diabetes (T2DM) (T2DM + Vit. D, *n* = 18; T2DM – Vit. D, *n* = 14) and patients without T2DM (Control + Vit. D, *n* = 30; Control − Vit. D, *n* = 30) before and after three-month supplementation. Results are presented as mean ± SD (* *p* < 0.05; *** *p* < 0.001).

**Figure 3 ijms-20-02891-f003:**
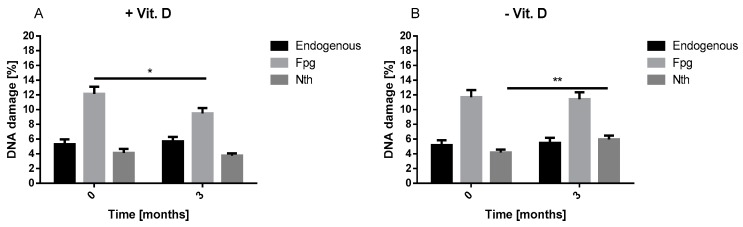
The impact of three-month supplementation of vitamin D on the level of endogenous and oxidative DNA damage in the intervention group (+Vit. D, *n* = 48) in relation to the comparative group (−Vit. D, *n* = 44). The level of DNA damage was measured by the alkaline comet assay. Data are expressed as mean ± SEM. * *p* < 0.05; ** *p* < 0.01.

**Figure 4 ijms-20-02891-f004:**
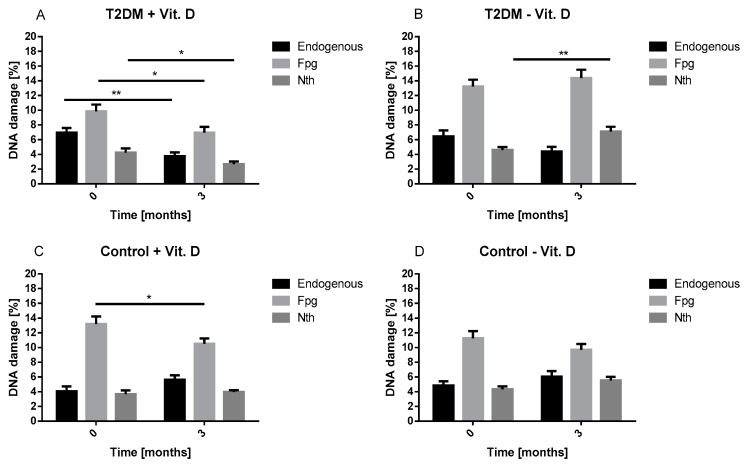
The impact of three-month supplementation with vitamin D on the level of endogenous and oxidative DNA damage. DNA damage were determined in (**A**) T2DM patients treated with vitamin D (T2DM + Vit. D, *n* = 14), (**B**) T2DM patients not treated with vitamin D (T2DM − Vit. D, *n* = 18), (**C**) patients without T2DM treated with vitamin D (Control + Vit.D, *n* = 30), (**D**) patients without T2DM not treated with vitamin D (Control − Vit. D, *n* = 30). The level of DNA damage was measured by the alkaline comet assay. Data are expressed as mean ± SEM. * *p* < 0.05; ** *p* < 0.01.

**Figure 5 ijms-20-02891-f005:**
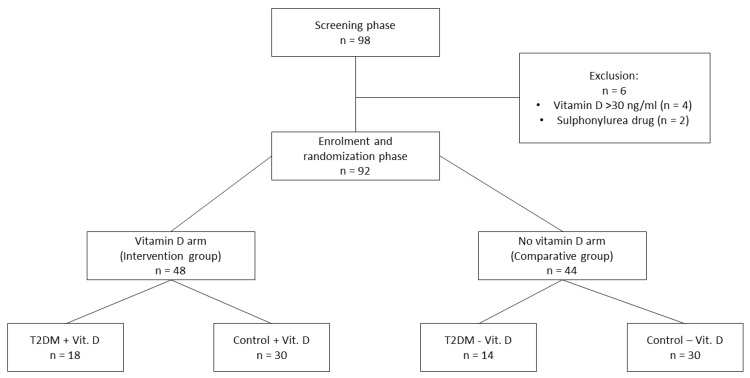
Patient flow diagram.

**Table 1 ijms-20-02891-t001:** Metabolic parameters of participants from the intervention group (*n* = 48) and the comparative group (*n* = 44) at baseline and after three-month vitamin D supplementation. Data are expressed as mean ± SEM.

Metabolic Parameters	Time (Months)	Intervention Group (+Vit. D) (*n* = 48)	Comparative Group (−Vit. D) (*n* = 44)
**FPG (mg/dL)**	0	113.63 ± 4.26	124.48 ± 4.70
	3	104.19 ± 3.06	105.15 ** ± 3.50
**TC (mg/dL)**	0	189.54 ± 7.94	162.12 ± 7.34 ^a^
	3	208.08 ± 8.23	173.93 ± 9.65 ^aa^
**HDL (mg/dL)**	0	46.60 ± 2.44	45.03 ± 3.14
	3	58.90 ± 1.88 ***	52.10 ± 2.22
**LDL (mg/dL)**	0	109.22 ± 7.28	92.65 ± 6.43
	3	123.08 ± 7.58	96.35 ± 7.63 ^a^
**TG (mg/dL)**	0	149.12 ± 12.64	117.15 ± 8.56 ^a^
	3	138.80 ± 7.64	130.77 ± 9.01
**HbA1c (%)**	0	6.64 ± 0.20	6.22 ± 0.13
	3	6.35 ± 0.21	6.42 ± 0.18
**HOMA-IR**	0	5.55 ± 0.59	7.85 ± 0.89 ^a^
	3	3.23 ± 0.38 **	4.49 ± 0.55 **
**TG/HDL ratio**	0	3.91 ± 0.44	3.69 ± 0.64
	3	2.51 ± 0.21 **	2.94 ± 0.43
**BMI**	0	28.12 ± 0.57	28.69 ± 0.39
	3	27.79 ± 0.53	28.13 ± 0.37

** *p* < 0.01; *** *p* < 0.001 Time (0 month) vs. Time (three months). ^a^
*p* < 0.05; ^aa^
*p* < 0.01 + Vit. D vs. − Vit. D. Grey colour indicates statistically significant differences between metabolic parameters.

**Table 2 ijms-20-02891-t002:** Metabolic parameters of patients with T2DM (T2DM + Vit. D, *n* = 18; T2DM – Vit. D, *n* = 14) and patients without T2DM (Control + Vit. D, *n* = 30; Control − Vit. D, *n* = 30) at baseline (0) and after three months of vitamin D supplementation (3). The FPG, TC, LDL, HDL, TG and HbA1C were evaluated. Data are expressed as mean ± SEM * *p* < 0.05; ** *p* < 0.01; *** *p* < 0.001.

Metabolic Parameters	Time (Months)	T2DM + Vit.D (*n* = 18)	T2DM − Vit. D (*n* = 14)	Control + Vit. D (*n* = 30)	Control − Vit. D (*n* = 30)
**FPG (mg/dL)**	0	135.39 ± 5.43	141.35 ± 9.30	101.53 ± 4.75 ^bbb,dd^	116.60 ± 4.70 ^ccc,eee^
	3	118.15 ± 2.74 **	125.11 ± 7.01	95.81 ± 3.88 ^bbb,ddd^	95.83 ± 2.56 ***^,ccc,eee^
**TC (mg/dL)**	0	141.63 ± 6.59	141.64 ± 10.64	215.10 ± 9.12 ^bbb,ddd^	172.35 ± 9.03 ^c,e,ff^
	3	171.45 ± 6.94 **	154.90 ± 13.39	223.00 ± 8.27 ^bbb,ddd^	183.95 ± 7.04 ^e,fff^
**HDL (mg/dL)**	0	34.25 ± 2.48	37.64 ± 4.25	53.90 ± 2.88 ^bbb,dd^	48.48 ± 4.00 ^c^
	3	53.91 ± 2.65 ***	47.01 ± 4.09	62.83 ± 2.26 *^,b,ddd^	57.08 ± 2.43 ^e,f^
**LDL (mg/dL)**	0	78.69 ± 6.30	75.40 ± 6.88	125.50 ± 9.79 ^bb,dd^	103.97 ± 7.80 ^c,e^
	3	88.36 ± 5.63	75.54 ± 10.27	137.22 ± 7.74 ^bbb,ddd^	107.30 ± 5.37 ^c,ee,ff^
**TG (mg/dL)**	0	143.44 ± 10.97	149.00 ± 18.17	152.37 ± 19.10	102.29 ± 7.91 ^cc,ee,f^
	3	148.18 ± 12.54	162.40 ± 22.90	131.89 ± 9.22	116.01 ± 6.14 ^cc,ee^
**HbA1c (%)**	0	7.90 ± 0.36	7.09 ± 0.26	5.92 ± 0.07 ^bbb,ddd^	5.781 ± 0.07 ^ccc,eee^
	3	7.44 ± 0.36	7.25 ± 0.27	5.91 ± 0.07 ^bb,ddd^	5.99 ± 0.08 ^ccc,eee^
**HOMA-IR**	0	11.68 ± 2.21	8.51 ± 1.11	4.43 ± 0.70 ^bb,dd^	7.53 ± 1.20 ^f^
	3	7.49 ± 0.87 *	5.99 ± 0.95	2.45 ± 0.36 *^,bb,ddd^	3.79 ± 0.64 **
**TG/HDL ratio**	0	4.81 ± 0.60	5.99 ± 1.76	3.38 ± 0.58	2.61 ± 0.30 ^ccc,e^
	3	3.13 ± 0.37 *	4.47 ± 1.17	2.25 ± 0.21 ^b,d,^*	2.23 ± 0.21 ^c,e^
**BMI**	0	29.95 ± 0.88	28.30 ± 0.83	27.02 ± 0.66 ^bb^	28.88 ± 0.41 ^f^
	3	29.41 ± 0.78	28.27 ±0.81	26.82 ± 0.65 ^b^	28.06 ± 0.39

* *p* < 0.05; ** *p* < 0.01; *** *p* < 0.001 Time (0 month) vs. Time (three months). ^b^
*p* < 0.05; ^bb^
*p* < 0.01; ^bbb^
*p* < 0.001 T2DM + Vit. D vs. Control + Vit. D. ^c^
*p* < 0.05; ^cc^
*p* < 0.01; ^ccc^
*p* < 0.001 T2DM + Vit. D vs. Control – Vit. D. ^d^
*p* < 0.05; ^dd^
*p* < 0.01; ^ddd^
*p* < 0.001 T2DM – Vit. D vs. Control + Vit. D. ^e^
*p* < 0.05; ^ee^
*p* < 0.01; ^eee^
*p* < 0.001 T2DM – Vit. D vs. Control – Vit. D. ^f^
*p* < 0.05; ^ff^
*p* < 0.01; ^fff^
*p* < 0.001 Control + Vit. D vs. Control – Vit. D. Grey colour indicates metabolic parameters that are statistically significant between baseline and after 3 months.

**Table 3 ijms-20-02891-t003:** The correlation between vitamin D and endogenous and oxidative DNA damage in the intervention group (+Vit. D) and comparative group (−Vit. D) at baseline (0) and after three months of vitamin D supplementation (3). The analysis was performed by Pearson’s correlation test.

DNA Damage	Time (Months)	Intervention Group (+Vit. D, *n* = 48)		Comparative Group (−Vit. D, *n* = 44)	
		Pearson Correlation *r*	*p*	Pearson Correlation *r*	*p*
**Endogenous**	0	0.081	0.582	−0.015	0.939
	3	−0.083	0.575	−0.110	0.563
**Fpg**	0	−0.069	0.640	0.275	0.705
	3	−0.215	0.140	−0.195	0.204
**Nth**	0	0.187	0.202	−0.226	0.141
	3	−0.278	0.06	−0.094	0.545

**Table 4 ijms-20-02891-t004:** The correlation between Vitamin D and endogenous and oxidative DNA damage in patients with T2DM (T2DM + Vit. D, *n* = 18; T2DM – Vit. D, *n* = 14) and patients without T2DM (Control + Vit. D, *n* = 30; Control – Vit. D, *n* = 30) at baseline (0) and after three months of vitamin D supplementation (3). The analysis was performed by Pearson’s correlation test.

DNA Damage	Time (Months)	T2DM + Vit. D *n* = 18		T2DM − Vit. D *n* = 14		Control + Vit. D *n* = 30		Control − Vit. D *n* = 30	
		Pearson Correlation *r*	*p*	Pearson Correlation *r*	*p*	Pearson Correlation *r*	*p*	Pearson Correlation *r*	*p*
**Endo-genous**	0	0.267	0.344	−0.487	0.077	0.299	0.471	−0.015	0.939
	3	0.458	0.056	−0.383	0.176	−0.028	0.883	−0.110	0.563
**Fpg**	0	−0.101	0.690	0.094	0.749	−0.102	0.592	0.161	0.395
	3	−0.244	0.328	−0.391	0.166	−0.237	0.206	−0.134	0.479
**Nth**	0	0.129	0.610	−0.267	0.356	0.275	0.141	−0.995	0.601
	3	0.060	0.814	−0.231	0.427	−0.245	0.191	−0.122	0.518

**Table 5 ijms-20-02891-t005:** The correlation between HbA1c and endogenous and oxidative DNA damage in patients with T2DM (T2DM + Vit. D and T2DM − Vit. D) at baseline and after three months of vitamin D supplementation. The study was perfomed by Pearson’s correlation test, * denotes statistically significant correlation

DNA Damage	Time (Months)	T2DM + Vit. D *n* = 18		T2DM – Vit. D *n* = 14	
		Pearson Correlation *r*	*p*	Pearson Correlation *r*	*p*
**Endogenous**	0	0.302	0.220	0.015	0.960
	3	−0.406	0.617	0.165	0.571
**Fpg**	0	0.126	0.620	0.229	0.430
	3	0.107	0.670	0.174	0.550
**Nth**	0	0.498	0.031*	−0.022	0.941
	3	0.194	0.440	−0.33	0.260

* indicates statistically significant correlation.

**Table 6 ijms-20-02891-t006:** The correlations between Vitamin D and metabolic parameters in the intervention group and comparative group at baseline and after three months of the study. The Pearson’s correlation test was employed, * denotes statistically significant correlation.

Metabolic Parameters	Time (Months)	Intervention Group (+Vit. D, *n* = 48)		Comparative Group (−Vit. D, *n* = 44)	
		Pearson Correlation *r*	*p*	Pearson Correlation *r*	*p*
**FPG (mg/dL)**	0	−0.131	0.362	−0.051	0.750
	3	0.120	0.419	0.097	0.548
**TC (mg/dL)**	0	0.264	0.069	0.197	0.199
	3	0.033	0.821	0.077	0.617
**HDL (mg/dL)**	0	0.299	0.038 *	0.191	0.220
	3	−0.030	0.837	0.134	0.384
**LDL (mg/dL)**	0	0.241	0.098	0.169	0.273
	3	0.053	0.717	−0.014	0.930
**TG (mg/dL)**	0	−0.088	0.551	−0.076	0.623
	3	−0.052	0.723	0.120	0.436
**HbA1c (%)**	0	−0.176	0.229	0.077	0.620
	3	−0.290	0.045 *	0.234	0.126
**HOMA-IR**	0	−0.220	0.132	−0.031	0.840
	3	0.187	0.231	0.060	0.700
**TG/HDL ratio**	0	−0.240	0.104	−0.270	0.071
	3	−0.120	0.401	−0.130	0.380
**BMI**	0	−0.087	0.555	−0.064	0.682
	3	−0.038	0.798	0.030	0.847

* denotes statistically significant correlation.

**Table 7 ijms-20-02891-t007:** The correlation between vitamin D and FPG, HbA1C, TC, LDL, HDL, TG in patients with T2DM (T2DM + Vit. D, *n* = 18; T2DM – Vit D, *n* = 14) and patients without T2DM (Control + Vit. D, *n* = 30; Control – Vit. D, *n* = 30) at baseline and after three months of the study. The analysis was conducted by Pearsons Correlation test, * denotes statistically significant correlation.

Metabolic Parameters	Time (Months)	T2DM + Vit. D *n* = 18		T2DM − Vit. D *n* = 14		Control + Vit. D *n* = 30		Control − Vit. D *n* = 30	
		Pearson Correlation *r*	*p*	Pearson Correlation *r*	*p*	Pearson Correlation *r*	*p*	Pearson Correlation *r*	*p*
**FPG (mg/dL)**	0	−0.073	0.776	0.311	0.274	−0.011	0.938	−0.380	0.038 *
	3	−0.031	0.917	−0.087	0.765	0.199	0.290	−0.122	0.518
**TC (mg/dL)**	0	0.143	0.570	0.263	0.364	0.190	0.313	0.217	0.250
	3	0.030	0.903	0.162	0.580	0.052	0.785	0.155	0.411
**HDL (mg/dL)**	0	0.341	0.165	0.325	0.256	0.211	0.263	0.237	0.206
	3	0.194	0.440	0.211	0.467	−0.162	0.394	0.231	0.219
**LDL (mg/dL)**	0	0.184	0.464	0.247	0.394	0.175	0.354	0.137	0.468
	3	−0.005	0.982	0.122	0.677	0.099	0.602	0.084	0.656
**TG (mg/dL)**	0	−0.484	0.042 *	−0.180	0.538	−0.045	0.810	−0.051	0.788
	3	−0.143	0.571	−0.002	0.993	0.007	0.969	−0.009	0.960
**HbA1c (%)**	0	−0.169	0.501	0.158	0.589	-	-	-	-
	3	−0.709	0.0009 *	0.317	0.269	-	-	-	-
**HOMA-IR**	0	−0.421	0.080	0.311	0.281	−0.091	0.631	−0.160	0.388
	3	−0.052	0.899	0.161	0.590	0.428	0.024 *	−0.149	0.438
**TG/HDL ratio**	0	−0.470	0.047 *	−0.390	0.162	−0.144	0.483	−0.332	0.081
	3	−0.340	0.171	−0.354	0.228	0.061	0.775	0.084	0.662
**BMI**	0	−0.015	0.954	−0.323	0.261	−0.270	0.150	−0.156	0.410
	3	−0.411	0.090	−0.293	0.309	−0.427	0.019 *	0.012	0.949

* indicates statistically significant correlation.

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
