# Peer review of "Vitamin D Supplementation Reduces Both Oxidative DNA Damage and Insulin Resistance in the Elderly with Metabolic Disorders"

_ijms, 2019, doi:10.3390/ijms20122891_

Round 1
Reviewer 1 Report
1. The present work largely describes largely not novel results, given that the association of vitamin D deficiency with diabetes and the beneficial effects of vitamin D supplementation on metabolic parameters related to glycaemic and lipidic homeostasis have been already reported by several authors. Orignal results, that are the most interesting here shown, are those related to beneficial effects of vitamin D supplementation on oxidative stress-related DNA damage.
In general comments and discussion of results obtained from the analysis of metabolic parameters are confused and a bit hard to understand.
Several issues have to be addressed.
2. How the authors explain the increase of serum vitamin D levels in not supplemented group after 3-months (Fig. 2,3), and the significantly higher levels of vitamin D in the T2M-vitD group compared with the control-vit D group?
3. The authors show that vitamin D supplementation increases HDL levels in T2M group. Therefore, how they explain that LDL serum levels are increased instead of being reduced, and the increase of TC after three months is higher than the sum of increased HDL and LDL?
4. The authors should comment about the reduction of TG/HDL ratio in all groups after three months, independently from vitamin D supplementation.
5. The treatment protocol involves a supplementation with 2000 IU/day of vitamin D, a dose that seems very high. The authors should comment on possible toxic effects of vitamin D excess intake that are already reported.
The design of the study should exclude aspecific effects deriving from amelioraton of life style in control group not supplemented with vitamin D. The authors must acknowledge these limitations.
Author Response
1. Thank you for this remark. According to your suggestion we modified the order of presented results in our paper as well as added key information connected with diabetes/hyperglycemia related oxidative stress and insulin resistance to the Introduction (page 2, line 48) and Discussion (page 12, line 364).
2. As for the first part of the question, in the group without vitamin D supplementation, after 3 months the concentration of vitamin D was increased from 15.58±1.25 to19.80±2.25 ng/ml. This result is not statistically significant. It should be noted that the higher SD value after 3 months of supplementation indicates a larger distribution of results around the average. This may indicate that some of the subjects consumed rich
vitamin D which allowed for a slight increase in vitamin D concentration. For comparison in groups with vitamin D supplementation was increase from 15.99±1.11 to 31.81±1.45. Responding to the second part of the question, it should be noted that T2DM-Vit. D group included 18 people in comparison with 30 people in the Control - Vit. D . After careful analysis of the data it cannot be excluded that in T2DM-Vit. D group few participants achieved a physiological vitamin D concentration which might result from the fact that they did not follow the instructions for not using vitamin D supplements and/or consuming food products rich in this vitamin.
3. In the study, LDL was not calculated directly but from the Friedewald formula. In this formula, the concentration of triglycerides is taken into account. In samples with high triglyceride content, the use of Friedewald's formula leads to lower LDL cholesterol concentrations, which can contribute to such results.
4. Thank you for this remark. The decrease in TG/HDL ratio was observed in all four groups, but in groups with vitamin D it was statistically significant. We introduced corresponding comment (page 7, line 249)
5. According to the recommendations of the Polish working group of International University Family Medicine Club , in people with vitamin D deficiency, the recommended therapeutic dose is 1000 ÷ 10,000 IU/day (about 50.000 IU/week) for 1-3 months. It should be also noted that Poland is a country with moderate climate and vitamin D deficiencies are common.
Reviewer 2 Report
Dear Editor,
Thank you very much for an opportunity to review the manuscript entitled: “Does Vitamin D have an impact on glucose and lipid 3 metabolism parameters and DNA damage in patients 4 with and without type 2 diabetes mellitus (T2DM)?” by Wenclewska at al. The manuscript is well written and contains some new and important observations concerning potential impact of vitamin D on glucose and lipids metabolism in type 2 diabetic patients. The manuscript could be accepted for publication in IJMS after some clarification.
Major comments:
1. First of all, it is not clear how the participants were selected. Point 2.2. clearly stated that inclusion criterion is T2DM, while only 32 from 92 participants had T2DM (figure 1). Please, clarify.
2. How would you explain statistically significant differences between potentially randomized groups (+/- vitamin D) before supplementation (time 0), Table 1,2, .
3. Did you check the effect of gender (women vs men)?
Minor comments
4. Please explain all abbreviations.. For example HOMA-IR
5. Please explain the criteria for HOMA-IR and TG/HDL ratio interpretation used in the study.
Author Response
1. Thank you for this comment. Our goal was to show the effect of vitamin D on metabolic parameters connected with insulin resistance. Please note, that not all patients with metabolic disorders/insulin resistance suffer from T2DM. In agreement, all participants of our study had metabolic disorders, but 1/3 additionaly suffered from T2DM. Therefore, to clarify our aim of the study we specified inclusion and exclusion criteria. in the section “Study Design” and added diagram presenting patients flow (page 2, line 74)
2. I would like to point out that there are no significant differences in the majority of the tested parameters. The significant differences between intervention and comparative group might result from the presence in groups patients with T2DM. If we look at the distribution of results at time 0 in all 4 groups (T2DM+Vit. D, T2DM- Vit. D, Control +Vit. D, Control-Vit. D) the groups remain similar.
3. Due to the low number of participants in subgroups with and without T2DM, we decided not to perform this analysis.
4.5. All abbreviations have been explained in the work.
Reviewer 3 Report
In this manuscript, the authors investigated the effect of vitamin D supplementation on metabolic parameters and DNA damage in individuals with or without type 2 diabetes, and reported that vitamin D supplementation significantly increased HDL cholesterol and decreased DNA damage. Although the interpretation of the study is limited due to a relatively small number of participants, the presented data seem intriguing. Since this manuscript may not be within the scope of International Journal of Molecular Sciences, I would recommend that it be submitted to a suitable journal.
[specific comments]
#English editing may be performed throughout the manuscript.
#References should be numbered consecutively and described precisely.
#1. (page 3, lines 109-112) FPG level was not significantly lower in Control + Vit. D compared with Control – Vit. D at baseline.
#2. (page 5, lines 179-185) Fig. 1 should be Fig. 2, and Fig. 2 should be Fig. 3.
#3. (page 6, line 200) HOMA-IR and TG/HDL ratio were decreased (not increased).
#4. (page 6, line 202) FPG and HOMA-IR were decreased (not increased).
#5. (page 10, line 299) I don’t understand why correlation analysis between vitamin D and HbA1c in patients without T2DM was not performed.
Author Response
1. Indeed, FPG level in Control+Vit.D group was not statistically lower than in the Control-Vit. D group. It was modified.
2. We corrected all figures’ numbers in the text. Please also note that we changed the order of presented results and in consequence the numbers of all figures and tables were modified.
3. We apologize for this mistake. We corrected this information in the text. (page 8, line 250)
4. We apologize for this mistake. We corrected this information in the text. (page 7, line 246)
5. HbA1c is a marker of metabolic control of diabetes. HbA1c level reflects the average glucose concentration from last 3 months. In case of patients suffering from diabetes who do not comply dialectologist’s recommendations, the value of HbA1c is increased, as a results of increased blood glucose levels. In turn, patients with normal daily glucose levels, the value of HbA1c is not higher than 6.4%. Therefore, in patients without diabetes we do not perform correlation analysis between vitamin D and HbA1c.
Round 2
Reviewer 1 Report
The Authors satisfactorily addressed all raised criticisms. The manuscript quality has been significantly improved
Author Response
Thank you for the acceptance of our answers and for your valuable remarks.
Reviewer 2 Report
Dear Editor,
Thank you very much for an opportunity to review of the manuscript entitled: “Vitamin D supplementation reduces both, oxidative 2 DNA damage and insulin resistance in the elderly 3 with metabolic disorders”. This is very interesting study by the manuscript requires some changes before publication.
Major comments:
1. The T2DM group is smaller than control 34 vs 60, with only 14 patients in T2DM – Vit D arm). It may affect the statistical evaluation of the results of study. For instance, 1 or 2 patients in T2DM – Vit D arm with significantly different 25(OH)D3 serum levels could be responsible for high variation in this group at time 0. Thus, I would suggest to increase number of T2DM patients in the study.
2. It has to be explained why you fail to show the effects of vitamin D supplementation (after 3 months) in T2DM patients. No statistical difference between T2DM – Vit D vs T2DM + Vit D arm. This observation actually decreases the significance of other results.
3. Please provide detailed methodology concerning FPG, HbA1c, TC, HDL, LDL, TG serum level measurements. Was it done in the commercial laboratory or you did it by yourself?
4. The discussion should be shortened and focused on main result. Also authors should underline that small T2DM group could have influenced the outcome of study thus interpretation of the data should be done with caution. I would use “results suggest” and “it requires further studies” rather than “indicate” or “can be stated”.
Minor point:
5. In the manuscript you use 2 abbreviations Fpg and FPG, which actually represent different tests but it could be confusing, please change.
6. How did you read the results of “DNA repair enzyme treatment” I think the last steps of procedure are missing.
7. Please make sure that all statistically significant differences (between bars) are shown on figures
8. Please explain “endogenous” (figures and table), there is no reference to “endogenous” in the legends nor in the text or the manuscript. Does it refer to commet assay?
9. Page 3, Line 101, Are you sure that T2DM patients were “diagnosed by the usage of hypoglycemic medication”?
10. Page 3. Line 114 “they are also had lower body weight” please correct
11. Page 12 Line 364 “vitamin D decreased oxidative DNA damage” many papers concerning the subject please search pubmed and cite.
Author Response
Thank you for your comments. We carefully considered all your remarks and modified the manuscript in accordance with you suggestions. We are convinced that the changes we made significantly improved the quality of our paper.
1.,2. We agree entirely with your comment. Indeed, we did not find statistically different 25(OH)D3 serum levels between T2DM – Vit D vs T2DM + Vit D after 3 months, but there still was a noticeable trend toward 25(OH)D3 increase in T2DM + vitD. As you suggested above, it could be the effect of a relatively small number of subjects in the T2DM group. However, please pay attention to the level of serum 25(OH)D3 in both T2DM groups at the baseline. Data for subjects in T2DM – Vit D was more scattered in comparison to other groups, as indicated by SD. It should be emphasized that in T2DM + vitD we observed a significant increase of level of 25(OH)D3 after 3 month supplementation - above 30ng/ml, whereas in T2DM –vit. D the level of 25(OH)D3 was insignificantly higher after 3 month than at time 0 but still below 30 ng/mL.We realize that these observations may decrease the significance of other results. However, please note that a short period of supplementation with 2000 IU of vitamin D increased the serum level of 25(OH)D3 to the level considered as physiological and was associated with significant reduction of HOMA-IR and TG/HDL ratio as well as an increase in HDL level, not only in T2DM groups.
We are aware of this weak point and mentioned it in our study’s limitations. Indeed, performing further studies including more participants with T2DM would reduce variations in the level of 25(OH)D3, and as a result increase statistical significance. However, please note that the primary aim of our study was to assess the effect of 2000 IU of vitamin D supplementation on DNA damage and metabolic parameters associated with insulin resistance in the population of people over 45 years with vitamin D deficiency (< 30 ng/ml). Due to the fact that the incidence of type 2 diabetes increases in this population, we decided to perform additional analysis in relation to T2DM presence/absence.
We would like to emphasize that the beneficial effect of vitamin D on DNA damage and metabolic parameters found in T2DM subgroup encouraged us to broaden our study. Namely, we intend to perform similar studies on T2DM patients involving a greater number of subject. We consider the prolongation of supplementation time and adjustment of vitamin D dose to the level detected at baseline.
Page 6,7, line 226-231
3. We introduced the following sentences: page 5 line 159-164 in accordance with your suggestion. All tests were performed in a commercial laboratory. HbA1c level was determined by turbidimetric immunoinhibition method, FPG level was photometrically measured in ultraviolet using enzymatic reactions catalysed by hexokinase. The enzymatic colorimetric test was quantified for TC, HDL and TG using Beckman Coulter AU analyzers. LDL was calculated using the Friedewald formula. Insulin concentration was determined by electrochemiluminescence test.
4. We shortened the discussion and changed “results indicate” into “results suggest” page 18, line 546, 549.
Minor comments:
5. Fpg is the commonly used short name for Formamidopyrimidine DNA glycosylase, a DNA repair enzyme, which was explained in the manuscript - page 6, line 189, while the abbreviation FPG (fasting plasma glucose) is the level of fasting glucose – page 1, line 27. Although Fpg and FPG abbreviations may be confusing, we would not want to change them because both abbreviations are widely used and recognized.
6. Thank you for your attention. We added the following sentence to the section: 2.4.2. DNA repair enzyme treatment:” To analyze the net value of DNA damage recognized by the enzymes, we subtracted the DNA damage observed in the absence of the enzymes from that measured in the presence of them. Page 6, line 203-205.
7. After carefully checking our results, we ensured that the statistical significance between bars are presented correctly.
8. Endogenous refers to basic DNA damage, not induced by any DNA damaging factor. It reflects the level of DNA damage in subject. It was described in page 5 line 174. According to your valid comment, to preserve homogeneity between manuscript text and figures and tables, we changed “basic” to “endogenous”.
9. Indeed, the use of the word “diagnosed” is unfortunate. The sentence “…or diagnosed by the usage of hypoglycemic medication before study begun” (page 4 line 119,120) was changed into : “… or were classified to T2DM group based on the usage of hypoglycemic medication before study begun.”
10. Of course, this is a grammatical error. We corrected it.
11. We searched Pubmed and found corresponding papers – new references No 19,20 were added page 15 line 430Reviewer 3 Report
The revised version of the manuscript seems to be much improved compared with the original version. However, I'm not sure whether this mauscript is within the scope of International Journal of Molecular Sciences.
[specific comment]
#1. (page 5, line 186) "Fig.2" should be "Fig.3".
Author Response
1. Thank you for your attention. We changed it.Round 3
Reviewer 2 Report
Thank you for all changes,